# Combination of Baseline LDH, Performance Status and Age as Integrated Algorithm to Identify Solid Tumor Patients with Higher Probability of Response to Anti PD-1 and PD-L1 Monoclonal Antibodies

**DOI:** 10.3390/cancers11020223

**Published:** 2019-02-14

**Authors:** Maria Silvia Cona, Mara Lecchi, Sara Cresta, Silvia Damian, Michele Del Vecchio, Andrea Necchi, Marta Maria Poggi, Daniele Raggi, Giovanni Randon, Raffaele Ratta, Diego Signorelli, Claudio Vernieri, Filippo de Braud, Paolo Verderio, Massimo Di Nicola

**Affiliations:** 1Medical Oncology Unit, Fondazione IRCCS, Istituto Nazionale dei Tumori di Milano, Via Giacomo Venezian 1, 20133 Milan, Italy; mariasilvia.cona@istitutotumori.mi.it (M.S.C.); sara.cresta@istitutotumori.mi.it (S.C.); silvia.damian@istitutotumori.mi.it (S.D.); michele.delvecchio@istitutotumori.mi.it (M.D.V.); andrea.necchi@istitutotumori.mi.it (A.N.); martamariapoggi@gmail.com (M.M.P.); daniele.raggi@istitutotumori.mi.it (D.R.); giovanni.randon@istitutotumori.mi.it (G.R.); raffaele.ratta@istitutotumori.mi.it (R.R.); diego.signorelli@istitutotumori.mi.it (D.S.); claudio.vernieri@istitutotumori.mi.it (C.V.); filippo.debraud@istitutotumori.mi.it (F.d.B.); 2Bioinformatics and Biostatistics Unit, Fondazione IRCCS, Istituto Nazionale dei Tumori di Milano, Via Giacomo Venezian 1, 20133 Milan, Italy; mara.lecchi@istitutotumori.mi.it (M.L.); paolo.verderio@istitutotumori.mi.it (P.V.); 3IFOM (Fondazione Istituto FIRC di Oncologia Molecolare), via Adamello 16, 20139 Milan, Italy; 4Department of Oncology and Hemato-Oncology, Universita’ degli Studi di Milano, 20122 Milan, Italy

**Keywords:** immune-checkpoint inhibitors, LDH, biomarkers

## Abstract

Predictive biomarkers of response to immune-checkpoint inhibitors (ICIs) are an urgent clinical need. The aim of this study is to identify manageable parameters to use in clinical practice to select patients with higher probability of response to ICIs. Two-hundred-and-seventy-one consecutive metastatic solid tumor patients, treated from 2013 until 2017 with anti- Programmed death-ligand 1 (PD-L1)/programmed cell death protein 1 (PD-1) ICIs, were evaluated for baseline lactate dehydrogenase (LDH) serum level, performance status (PS), age, neutrophil-lymphocyte ratio, type of immunotherapy, number of metastatic sites, histology, and sex. A training and validation set were used to build and test models, respectively. The variables’ effects were assessed through odds ratio estimates (OR) and area under the receive operating characteristic curves (AUC), from univariate and multivariate logistic regression models. A final multivariate model with LDH, age and PS showed significant ORs and an AUC of 0.771. Results were statistically validated and used to devise an Excel algorithm to calculate the patient’s response probabilities. We implemented an interactive Excel algorithm based on three variables (baseline LDH serum level, age and PS) which is able to provide a higher performance in response prediction to ICIs compared with LDH alone. This tool could be used in a real-life setting to identify ICIs in responding patients.

## 1. Introduction

In the era of immunotherapy, several biological and biochemical factors have been investigated as potential biomarkers of tumor response/resistance to immune-checkpoint inhibitors (ICIs). Select patients is an important clinical need in an attempt to offer them the best therapeutic workup, to avoid unnecessary side effects, and to optimize the use of economic resources. In order to identify a predictive tool of response to ICIs, we evaluated the available and manageable parameters that could ameliorate the selection of patients. In this context, lactate dehydrogenase (LDH) is a potentially interesting, cheap and easy-to-detect biomarker of response [1,2,3,4]. Indeed, serum LDH levels are an independent poor prognostic factor in several malignancies, including renal cell [5] and nasopharyngeal carcinoma [6], lymphomas [7], multiple myeloma [8], sarcomas [9] and lung cancer [10,11]. It also seems to be predictive of clinical outcomes in patients treated with anti-PD1 monoclonal antibodies (mAbs) [12]. For instance, it inversely correlates with the probability of achieving a tumor response in metastatic melanoma patients treated with anti- Cytotoxic T-Lymphocyte Antigen 4 (CTLA4) mAbs [13,14]. In previous studies, the link between serum LDH levels and poor patient prognosis has been generally attributed to the fact that high LDH levels reflect a high tumor burden, which is often associated with worse clinical outcomes. However, LDH is an enzyme that catalyses the conversion of pyruvate to lactate in highly glycolytic cancer cells, and its serum levels could be a proxy of tumor metabolic activity and not simply of tumor burden. Of note, recent studies have suggested that enhanced glycolytic activity in human malignancies is associated with an immunosuppressive environment, while glycolysis inhibition reduces tumor infiltration by immunosuppressor myeloid cells (MDSCs), stimulating the infiltration by cytotoxic lymphocytes [15].

Potential biomarkers of response/resistance to immunotherapy other than serum LDH, such as intratumor Programmed death-ligand 1 (PD-L1) expression, tumor microenvironment characteristics, tumor mutational load, mismatch-repair deficiency, and neutrophil-lymphocyte (N/L) ratio in peripheral blood, have been extensively investigated [16]. Unfortunately, no single parameter has been consistently associated with tumor response and clinical outcomes in all types of neoplasms; moreover, many of these biomarkers require specific analyses in tumor specimens, which are not always available. Therefore, cheap and easy-to-measure biochemical and clinical parameters could significantly help in the selection of patients more likely to benefit from ICIs, without increasing costs.

Here, we describe an algorithm based on baseline serum LDH levels, patient Performance Status (PS) and age, which could help clinicians to provide more accurate identification of patient candidates to ICIs.

## 2. Results

In our study, we enrolled 271 metastatic solid tumor patients treated at Fondazione IRCCS—Istituto Nazionale dei Tumori with anti PD-1 and anti PD-L1 mAbs from April 2013 to August 2017. Patients were evaluated for baseline LDH serum level, PS, age, N/L ratio, type of immunotherapy, number of metastatic sites, histology, and sex. Overall, the population was made up of 43.2% (117) lung cancer, 22.1% (60) melanoma and 34.7% (94) miscellaneous other solid tumors (1 anal, 1 hepatocellular carcinoma (HCC), 1 thyroid, 1 germ cell tumor, 2 gynecologic, 3 gastric, 5 head and neck (H&N), 4 colorectal, 5 sarcoma, 6 biliary tract, 6 mesothelioma, 26 renal, and 33 urothelial). Patient’s characteristics of training, validation and overall cohort are reported in Table 1; the number of metastatic sites is defined as the number of involved organs; the PS is evaluated through the Eastern Cooperative Oncology Group (ECOG) criteria [17] and dichotomized as 0 or ≥1. All responses were assessed by computed tomography. The categories of response consisted of: complete response (CR), partial response (PR), stable disease (SD), or disease progression (DP) as per RECIST (Response Evaluation Criteria in Solid Tumours) 1.1 criteria [18]. Disease control (DC) was defined as any CR, PR or SD. Overall, 150 (55.35%) patients achieved DC (6 CR, 59 PR, 85 SD) and 121 (44.65%) had DP as their best response.

The evaluation of LDH levels was performed in terms of percentage increase with respect to the upper limit of the specific normality range. This transformed variable was called *LDH normalized.* The median value of *LDH normalized* distribution was −27.7%, the lower quartile −38.48% and the upper −3.96%. A minimum extreme value was observed at −66.96% and a maximum at 954.79%. Figure 1 shows the *LDH normalized* distributions in the training and validation cohorts when considering all patients (Figure 1A) or according to the best response achieved with immunotherapy (Figure 1B).

For all the continuous variables considered in the logistic regression model, we found that a linear relationship between the log odds and their values was satisfied.

Univariate analysis was performed in 104 patients achieving DC and 83 patients undergoing DP (training set); of note, clinical response was significantly associated (*p* < 0.0001) with *LDH normalized*, with an odds ratio estimate (OR) equal to 0.792 for any LDH increment of 10%. We also found a significant positive association between age and tumor response, with an OR of 1.426 for any 10-year increment (*p*-value: 0.0093), and an inverse association between ECOG PS with an OR equal to 0.530 for 1 vs. 0 score (*p*-value: 0.0419) or N/L ratio with an OR equal to 0.899 (*p*-value: 0.014) and tumor response. Then, a logistic multivariate model was built by including these four variables, and a backward selection procedure was performed. Baseline LDH serum levels, age and PS were independently associated with the probability of responding to the treatment, with a statistically significant (*p* < 0.05) or borderline significant (*p*-value: 0.056 in the case of PS) association; therefore, they were retained in the final model. On the other hand, the N/L ratio was removed because it was not independently associated with the chance of responding (*p*-value: 0.529). The predictive capability of the final model was evaluated by generating a receive operating characteristic curve (ROC) and using as a pivotal statistic the area under the ROC curve (AUC). A satisfactory predictive capability [19] was observed, showing an AUC of 0.771 (95% Confidence Interval (CI): 0.701;0.842). The contribution of each variable of the final model to the predictive performance is graphically shown in Figure 2, and the differences between AUCs of the *LDH normalized* univariate model and the final one turned out to be significantly different to zero (difference: −0.0585; *p*-value: 0.0298; 95% CI: −0.111; −0.0057). By applying the training coefficients to the validation set, the model was statistically validated showing a significant AUC of 0.685 (95% CI: 0.569;0.801) (Figure 3). When the validated model was fitted to the totality of 150 patients achieving DC and 121 patients undergoing DP, the impact of these three variables on tumor response remained significant, as shown in Table 2; the overall AUC value, as well as the cross-validated one, were satisfactory (AUC: 0.737 95% CI: 0.675;0.798 and AUC: 0.718 95% CI: 0.654;0.781, respectively). Finally, we implemented an interactive Excel tool like that shown, for feasibility purposes, in the example of Table 3. By inserting for each patient: the upper limit of the normal reference range of the adopted kit for LDH quantification, the baseline LDH serum value, the ECOG PS score (as 0, 1, 2), and the age, it is possible to obtain the corresponding estimated probability of clinical response to ICIs.

Finally, we compared the performance (in terms of AUC) of the predictor built starting from the final model, to that derived from the only N/L ratio. As reported in Figure 4, the first classifier, with an AUC equal to 0.737 (95% CI: 0.675; 0.798), showed a higher predictive capability with respect to the N/L ratio classifier characterized by an AUC value of 0.645 (95% CI: 0.579; 0.711). In particular, the AUC values’ difference was statistically significant (*p*-value: 0.0220).

## 3. Discussion

The renewed interest for immunotherapy in the last years and the recent introduction of several ICIs in the clinical practice have redefined the therapeutic strategies of different solid tumors. The efficacy of the immunological approach was first proven in advanced melanoma with the anti CTLA-4 mAb Ipilimumab [20]. Thereafter, also anti PD-1/PD-L1 mAbs were tested against tumors that were classically considered to be poorly immunogenic and mostly unresponsive to immunotherapy, such as non-small cell lung cancer (NSCLC); however, these drugs demonstrated impressive and long-lasting anticancer activity in a minority of patients [21,22,23,24,25]. Unfortunately, despite the remarkable clinical efficacy and low toxicity of ICIs, the vast majority of patients with advanced solid cancers fail to achieve durable responses with ICIs. Therefore, predictive biomarkers of clinical benefit from ICIs are urgently needed in order to select patients with a higher probability of response, as well as to optimize the available economic resources. PD-L1 expression, tumor microenvironment (TME) features, mutational load, mismatch-repair deficiency, and N/L ratio in peripheral blood have been extensively investigated [26]. However, a universally recognized biomarker is not available, yet. For example, although high intratumor PD-L1 expression seems to be significantly associated with a better response to PD-1/PD-L1 blockade agents in several tumors [27], the spatial heterogeneity and dynamic changes of expression in the same tumor, together with the lack of reliable detection methods and definite cut-off values, actually limits its widespread use in clinical practice.

In contrast, measuring LDH serum level is a simple and low-cost evaluation that has already been proposed as a biomarker predictive of tumor response/resistance to regorafenib [28], temsirolimus [29], sorafenib [30,31], and anti CTLA-4 mAbs [12] in patients with colorectal, renal, pancreatic cancers, HCC, and melanoma, respectively.

LDH serum levels have been historically considered to reflect the total number of viable and biologically active cancer cells inside a tumor mass; therefore, its inverse association with patient prognosis and/or tumor response to chemotherapy was mainly attributed to the association between serum LDH levels and tumor burden. However, LDH is a metabolic enzyme that takes crucially part to the glycolytic pathway, which is aberrantly activated in several human cancers to fuel tumor bioenergetics and anabolic needs [32,33]. Of note, enhanced glycolysis in cancer masses leads to reduced glucose levels in the TME and, consequently, to glucose starvation in cells of the TME, including cytotoxic lymphocytes that mediate the antitumor immune response. This metabolic competition between cancer cells and immune cells for the use of glucose molecules may be one crucial mechanism through which malignant cells inhibit the activity of cytotoxic lymphocytes. Therefore, the observed inverse association between serum LDH levels and clinical benefit from ICIs could reflect an impairment of antitumor immunity in highly glycolytic, LDH-overexpressing malignancies. An alternative explanation for the link between tumor glycolytic activity and response to immunotherapy comes from a recently published preclinical study, where the inhibition of glycolysis in different tumor models was associated with reduced secretion of Granulocyte Colony-Stimulating Factor (G-CSF) and Granulocyte-Macrophage Colony-Stimulating Factor (GM-CSF) by cancer cells and lower intratumor infiltration by MDSCs, which restrain the activity of cytotoxic lymphocytes [15]. Therefore, high serum LDH, which reflects tumor glycolytic activity, may also reflect a more immunosuppressive, MDSC-enriched tumor microenvironment.

In order to improve the predictive capability of the LDH serum level, we combined it with other clinical parameters for which a rationale exists to test them as predictive biomarkers. To this aim, we have created an interactive Excel tool based on three variables (baseline LDH, age and PS ECOG score) which is able to provide high accuracy in response prediction to ICIs if compared with LDH alone. Our retrospective analysis confirms that patients with high baseline LDH serum levels have a statistically significant reduced probability of achieving a clinical response during treatment with ICIs, especially in patients who are younger and have poorer performance status.

Since the N/L ratio, which reflects systemic cancer-related inflammatory status, has been proposed as a biomarker of resistance to chemo-immunotherapy, we decided to evaluate its impact on the patient’s response to ICIs [34]. We observed that the N/L ratio has a significant predictive role but only in univariate fashion and shows a worse predictive capability than our model. To assess the performance of our algorithm, we tried to test it in three of the four colorectal cancer tissues of whom microsatellite instability (MSI) has been previously evaluated: two out of the three patients scored MSI. It is well known that the MSI subset of colorectal cancer has a greater likelihood of response to ICIs compared to the stable one [35]. It is worth noting that our predictor was able to detect with high accuracy which one between the two MSI patients had a major probability of DC and can really benefit from ICIs (Table 4).

The algorithm that we developed and validated in this study could provide a base to guide physicians in a real-life setting to better plan a therapeutic strategy tailored to patient characteristics and potentially able to identify patients more likely to benefit from ICIs. The clinical relevance of our findings is related to the easy detectability and manageability of the variables tested. Indeed, information about age, baseline LDH serum levels and PS can be collected quickly, already during the medical examination.

The main limits of our study consist in its retrospective nature, the heterogeneity of tumor histologies included, and the relatively small number of patients enrolled. Prospective studies in larger populations and focused on specific tumor types have already started in order to validate our results.

## 4. Materials and Methods

### 4.1. Ldh Evaluation

The evaluation of LDH was performed at baseline. All measures were performed in our laboratory, with COBAS^®^ 6000 analyser (Roche Diagnostics, Indianapolis, IN, USA), using a UV-test. The catalytic activity of LDH was determined by the measurement of decreased absorbancy of nicotinamide adenine dinucleotide at 340 nm as a result of catalytic reduction of pyruvate to lactate. From January 2013 to July 2015, the normality reference range was 230–460 Units per liter (U/L); since August 2015, it has changed to 20–480 U/L.

### 4.2. Statistical Analysis

Discrete variables (line and type of therapy (anti-PD1 vs. anti-PDL1), number of metastatic sites, histology, and sex) were opportunely categorized by taking into consideration their clinical function and according to their distributions. Concerning the continuous variables, age and N/L ratio were used in their original scale, whereas for LDH, an appropriate transformation was applied to the original values in order to normalize levels determined by the UV-test with the two different normal reference ranges. All the analyses were performed in terms of percentage increase with respect to the upper limit of each specific range (*LDH normalized*). In order to investigate the relationship between the clinical response and continuous variables and to detect possible nonlinear effects, we resorted to a logistic regression model based on restricted cubic splines. A training set, consisting of all patients treated before the 1st of January 2016 (≈ 70% of all patients), was used to build models which were tested on a validation set including patients treated since the 1st of January 2016 (≈ 30% of all patients). The relationships between each variable and the clinical response (DC vs. DP) were investigated by resorting to a logistic regression model in both univariate and multivariate fashion. The hypothesis of OR equal to 1 was tested using the Wald Statistic. All the variables resulted statistically significant (α = 0.05) in univariate analysis was considered in the initial model of multivariate analysis, and a backward selection procedure was used to obtain the final model. We investigated the predictive capability of the multivariate model by means of the AUC. The nonparametric approach of DeLong and Clarke–Pearson [36] was used to compare the discriminatory performance of different models and evaluate the contribution of each variable of the model. The most satisfactory model was applied on the validation set to statistically validate it and it was fitted overall to obtain the most robust estimates. AUC estimates based on cross-validated predicted probabilities were determined to evaluate the performance of the selected variables in the absence of an independent dataset [37]. All statistical analyses were carried out with SAS software (Version 9.4.; SAS Institute, Inc., Cary, NC, USA) by adopting a significance level of α = 0.05. The overall coefficients estimated were used to implement an Excel algorithm that requires the selected clinical variables of the patient and returns the corresponding response probability.

## 5. Conclusions

Identifying responder patients before starting immunotherapy is an important clinical need in order to define the best therapeutic workup, avoid unnecessary side effects and efficiently use the economic resources. The clinical relevance of our findings is related to the easy detectability and manageability of the variables tested. Indeed, information about age, baseline LDH serum levels and PS can be collected quickly, already during the medical examination.

The main limits of our study consist in its retrospective nature, the heterogeneity of tumor histologies included, and the relatively small number of patients enrolled. Prospective studies in larger populations and focused on specific tumor types have already started in order to validate our results.

## Figures and Tables

**Figure 1 cancers-11-00223-f001:**
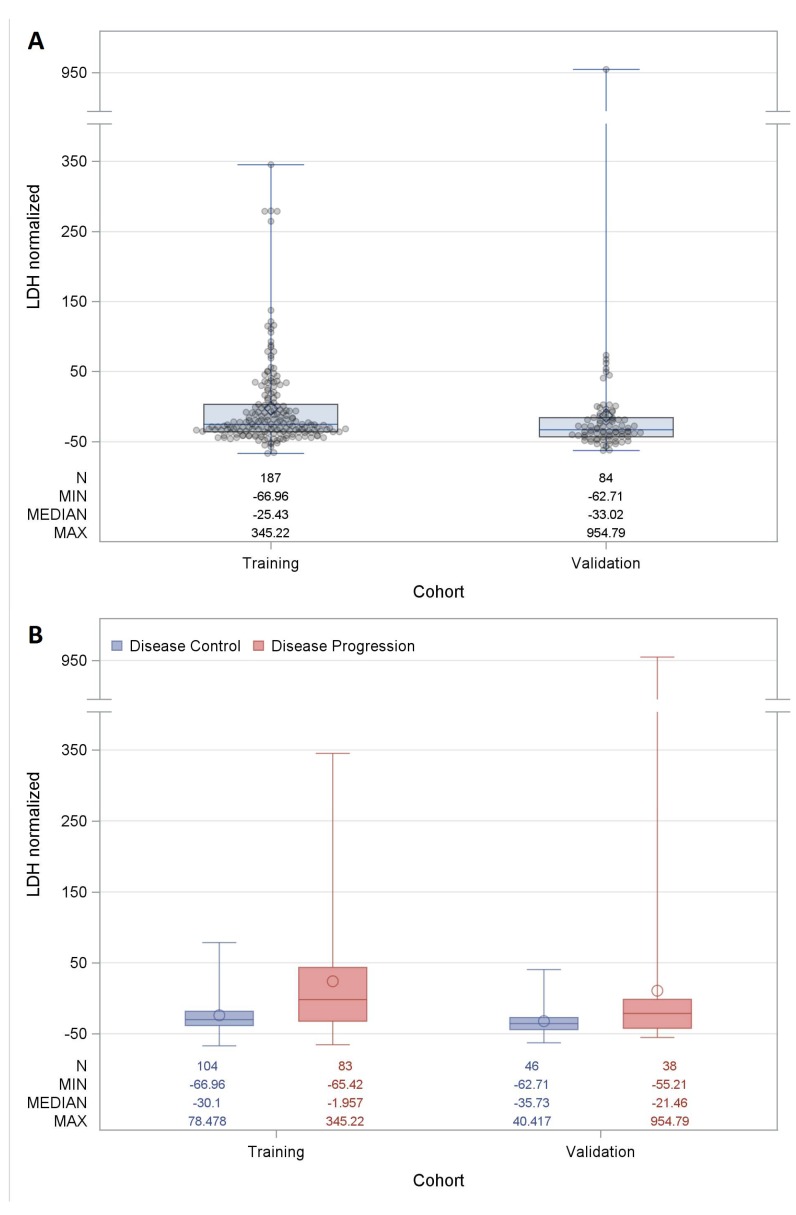
Distribution of lactate dehydrogenase (*LDH*) *normalized* in the training and validation cohort. (**A**) Boxplots reflecting the distribution of *LDH normalized* for each patient (*n* = 271) distinguished in training and in the validation cohort. Each box indicates the 25th and 75th centiles. The horizontal line inside the box indicates the median, and the whiskers indicate the extreme measured values. Each observation is represented by a grey dot. (**B**) Boxplots reflecting the distribution of *LDH normalized* according to the best response distinguished in training and in the validation cohort. Each box indicates the 25th and 75th centiles. Blue and red colors indicate disease control and disease progression patients, respectively. The horizontal line inside the box indicates the median, and the whiskers indicate the extreme measured values.

**Figure 2 cancers-11-00223-f002:**
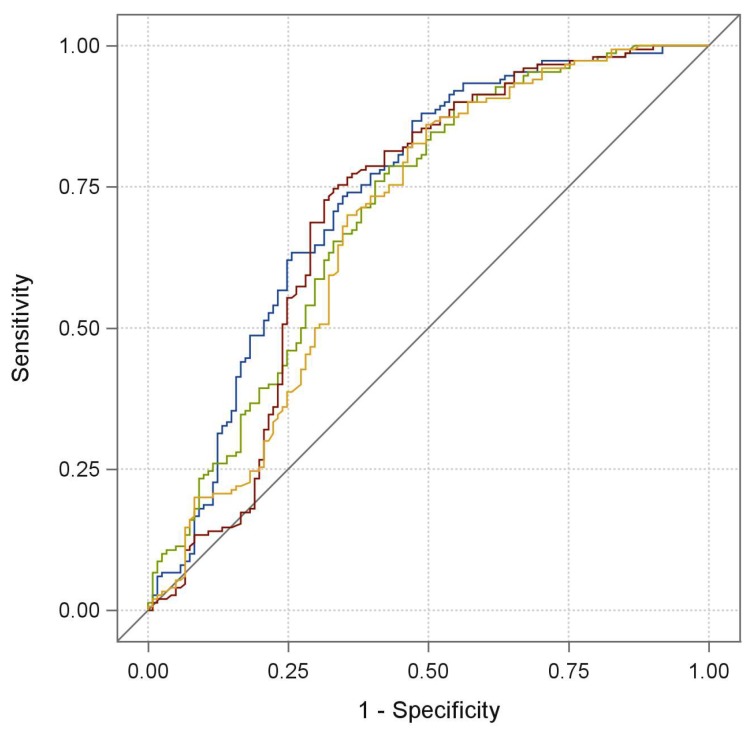
Receive Operating Characteristic (ROC) curves in the training set of the final multivariate model (blue line, Area under the ROC curve (AUC): 0.771), final model without performance status (green line, AUC: 0.749), final model without age (red line, AUC: 0.728), and *LDH normalized* univariate model (yellow line, AUC: 0.713).

**Figure 3 cancers-11-00223-f003:**
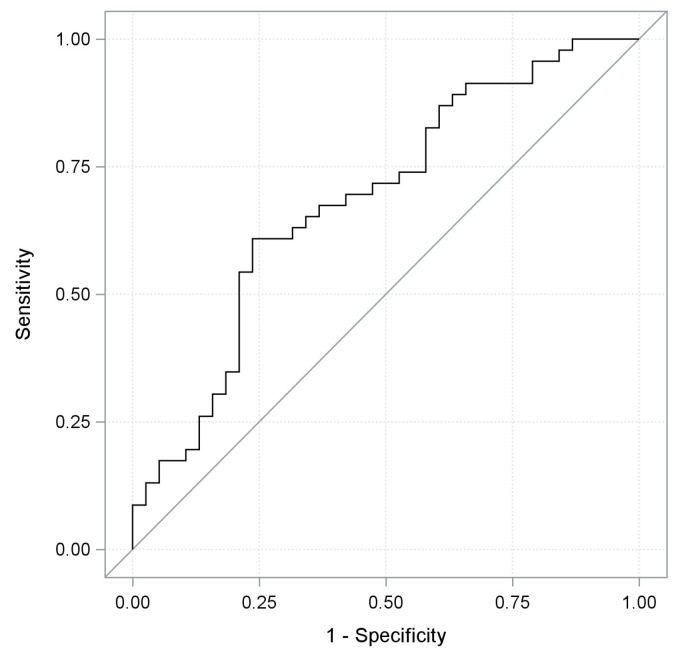
ROC curve of the final multivariate model applied on the validation set with an AUC value of 0.685.

**Figure 4 cancers-11-00223-f004:**
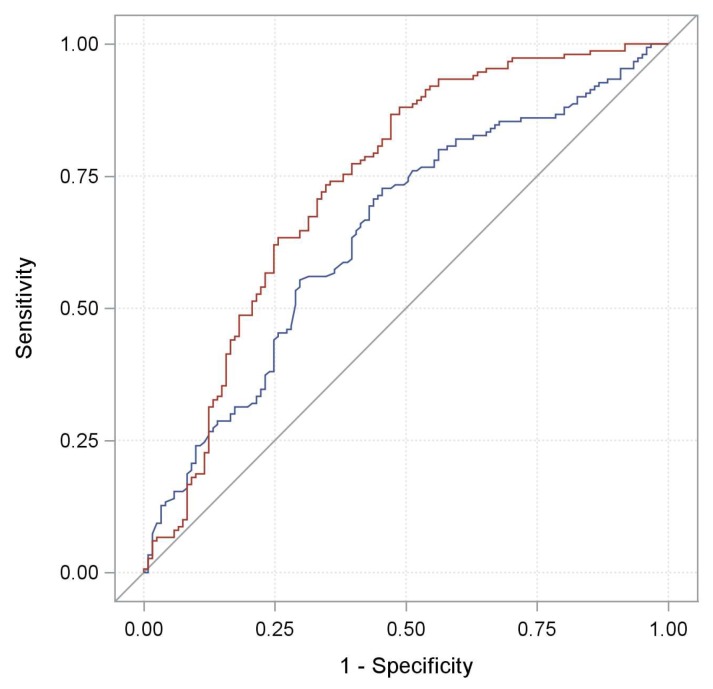
ROC curves of the proposed predictor (red line, AUC: 0.737) and N/L ratio one (blue line, AUC: 0.645).

**Table 1 cancers-11-00223-t001:** Clinic-pathological characteristics: training, validation and total cohorts.

Characteristics	Training Cohort		Validation Cohort		Total Cohort	
Categorical Variables	Freq	%	Freq	%	Freq	%
**Sex**						
Female	78	41.71	32	38.1	110	40.59
Male	109	58.29	52	61.9	161	59.41
**Tumor Site**						
Melanoma	36	19.25	24	28.57	60	22.14
Lung	93	49.73	24	28.57	117	43.17
Others *	58	31.02	36	42.86	94	34.69
**Treatment**						
PD-1	117	62.57	51	60.71	168	61.99
PD-L1	70	37.43	33	39.29	103	38.01
**Line of therapy**						
1	8	4.3	37	44.0	45	16.61
2	56	29.9	33	39.3	89	32.84
≥3	123	65.8	14	16.7	137	50.55
**Number of Metastatic sites**						
1	31	16.6	11	13.1	42	15.5
2	83	44.4	37	44.0	120	44.28
≥3	73	39.0	36	42.9	109	40.22
**PS (ECOG)**						
0	123	65.78	54	64.29	177	65.31
≥1	64	34.22	30	35.71	94	34.69
**Best Response**						
DC	104	55.61	46	54.76	150	55.35
DP	83	44.39	38	45.24	121	44.65
**Continuos variables**	**Median**	**Range**	**Median**	**Range**	**Median**	**Range**
**Age, years**	61	16; 84	66	34; 83	62	16; 84
**N/L ratio**	3.44	0.65; 39.50	3.39	0.78; 28.33	3.44	0.65; 39.50
**LDH serum level**	353.00	152.00; 2048.00	321.50	179.00; 5063.00	343.00	152.00; 5063.00

PS: performance status, DC: disease control, DP: disease progression, N/L ratio: neutrophil to lymphocyte ratio. * Other solid tumors: 1 anal, 1 HCC, 1 thyroid, 1 germ cell tumor, 2 gynecologic, 3 gastric, 5 H&N, 4 colorectal, 5 sarcoma, 6 biliary tract, 6 mesothelioma, 26 renal, and 33 urothelial.

**Table 2 cancers-11-00223-t002:** Overall Odd Ratio (OR) estimates and 95% Confidence Interval (CI) for each variable of the final model.

Effect	OR	95% CI	
LDH normalized for a 10% increment	0.810	0.744	0.883
Age for a ten-years increment	1.305	1.038	1.641
PS (ECOG) 1 vs. 0 score	0.481	0.274	0.846

OR: Odd Ratio; CI: Confidence Interval; PS: performance status; ECOG: Eastern Cooperative Oncology Group criteria.

**Table 3 cancers-11-00223-t003:** Example of the excel interactive tool. Grey cells need to be filled; the blue one will show the estimated probability of clinical response.

Variable	Value
**Kit Characteristic**	
Upper limit of normal reference range	460
**Patients Characteristics**	
LDH serum value	77
ECOG PS score [17]	1
Age	60
**Estimated Probability %**	76.39

**Table 4 cancers-11-00223-t004:** Estimated probability of response in colorectal patients according to microsatellite mutational status.

PS (ECOG)	Baseline LDH	Age (years)	Estimated Probability of Response (%)	Best Response	Microsatellite Instability
0	1063	60	8.21	DP	Yes
0	354	67	69.45	SD	Yes
1	925	47	5.29	DP	No

PS: performance status, DP: disease progression, SD: stable disease.

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
