# Peer review of "Combination of Baseline LDH, Performance Status and Age as Integrated Algorithm to Identify Solid Tumor Patients with Higher Probability of Response to Anti PD-1 and PD-L1 Monoclonal Antibodies"

_cancers, 2019, doi:10.3390/cancers11020223_

Round 1

Reviewer 1 Report

The authors propose an interactive Excel algorithm based upon baseline LDH, age and ECOG performance status.  Utilizing data collecting from 271 consecutive metastatic solid tumor patients treated from 2013 until 2017 with checkpoint inhibitors, they evaluated the variables with training and validation sets.  The authors propose utilizing this to potentially predict response to checkpoint inhibitors.  Biomarker development for predicting response to immunotherapy is a hot field. 

1.  Please describe in further detail the utilization of this algorithm in clinical practice.
2.  There are several different histologies utilized in the training and validation set.  If this were applied to a microsatellite stable colon cancer, what would be the results?
3.  Were there any specific variables looked at that pertained specifically to the immune system such as neutrophil to lymphocyte ratios.

4. Please describe in further detail the limitations of the study.

Author Response

1.      Please describe in further detail the utilization of this algorithm in clinical practice.

Identifying responder patients before starting immunotherapy is an important clinical need in order to define the best therapeutic workup, avoid unnecessary side effects and well use the economic resources. Our algorithm can be a practical and useful tool, because it’s based on easy detectable variables. Information about age, LDH and performance status can be collected quickly, already during the medical examination, and the Excel algorithm is intuitive and easy to fill.

Even if our data are very promising, they come from a retrospective study with a relatively small and heterogeneous population. In order to validate our results, a prospective study with a larger population for each histology is ongoing.

We addressed this issue in the text (Discussion Section)

2.  There are several different histologies utilized in the training and validation set.  If this were applied to a microsatellite stable colon cancer, what would be the results?

We are very thank for this suggestion. In our series of patients, only 3 colorectal patients were investigated for tumor microsatellite instability (MSI) status and 2 out of 3 scored MSI. Appling our algorithm to these cases, we observed a good performance to predict which MSI patient benefit from immunotherapy. However, a larger number of patients are needed to confirm this data. This comment has been added in Discussion section.

3.  Were there any specific variables looked at that pertained specifically to the immune system such as neutrophil to lymphocyte ratios.

According your suggestion, we evaluated the neutrophil/lymphocyte ratio of all 271 patients in an univariate fashion and we found a statistically significant association with tumor response, with an OR equal to 0.899 (p-value 0.014). By considering the N/L ratio in the initial multivariate model together with the other three significant variables (ECOG, LDH normalized and age) the multivariate p-value resulted equal to 0.529. Thus, through a backward selection procedure, this covariate was removed from the initial model. In addition, we compared the predictive capability of the two predictors, the one built starting from the final model and that derived from the N/L ratio, which showed an AUC of 0.737 (IC 95% 0.675-0.798) and 0.645 (IC 95% 0.579-0.711), respectively.  We integrated these new results in the revised manuscript.

4. Please describe in further detail the limitations of the study.

     The retrospective nature of the analysis, the heterogeneity of population and the relatively small number of patients are the principal limits of our study. For these reasons we have designed a prospective study in larger patient population for each tumor type in order to validate our results.

Reviewer 2 Report

In this manuscripts authors have attempted to develop a statistical method to predict/identify patients who would respond best to the ICIs. It is a nice attempt by the authors to utilize common/simple parameters like baseline LDH, age and ECOG score to identify the best responders to ICIs. However, it seems that more data would be needed to support the conclusions made in the manuscript. 

Also, it is not clear from the presented data that the described method would help the clinicians in deciding whether to go forward with the ICI therapy or not. It is a big decision to make for a clinician considering that it would have huge impact on the outcome of patient health. That is why it is imperative that further data is provided to validate the method described in this manuscript.

Introduction/background: See line 42, "..... metabolic switch from......., even in hypoxic condition (warburgh effect)." This statement is in accurate. Warburg effect is the increased/preferential use of glycolytic pathways by cancer cells even in the presence of oxygen. Authors should be careful while presenting the published facts. Also, authors should include more background information such as; statistics on patients who do not respond to ICIs, status of LDH in those patients, does the levels of LDH decides the outcome of ICI treatment, what is ECOG score? and why is it important. Including more background would help the readers to understand the significance of the work presented in this manuscript and could bolster authors' work.

Abbreviations such as DC, PD, ECOG, etc should be spelled at their first use. Especially, what is ECOG and why is it an important parameter? Authors' method has three parameters and one of them is ECOG however no where the authors have defined the meaning and significance of ECOG.

ROC has not been defined/spelled.

In the study population were the values used for LDH and ECOG were recorded before the patients were given ICIs or after or during the treatment. And who would that affect the conclusion made in the manuscript.

Author Response

It is not clear from the presented data that the described method would help the clinicians in deciding whether to go forward with the ICI therapy or not. It is a big decision to make for a clinician considering that it would have huge impact on the outcome of patient health. That is why it is imperative that further data is provided to validate the method described in this manuscript. 

Your comment is absolutely correct: in fact, to validate the data emerging from this first exploratory analysis, we have already planned a prospective study in larger and more homogeneous populations of cancer patients.

Introduction/background: See line 42, "..... metabolic switch from......., even in hypoxic condition (warburgh effect)." This statement is in accurate. Warburg effect is the increased/preferential use of glycolytic pathways by cancer cells even in the presence of oxygen. Authors should be careful while presenting the published facts.  

Thank you for your suggestion. Based on your comment, we have significantly modified the Introducton section. In the revised version of the introduction, we have discussed recent preclinical evidence showing a potential competition for glucose molecules in tumor microenvironment between cancer cells and antitumor T lymphocytes. High serum LDH levels in patients could reflect more glycolytic malignancies that undergo enhanced Warburg effect, while reducing glucose availability to infiltrating lymphocytes and contributing to hamper the antitumor immune response during treatment with ICIs.

Also, authors should include more background information such as; statistics on patients who do not respond to ICIs, status of LDH in those patients, does the levels of LDH decides the outcome of ICI treatment 

We thank the reviewer for his/her suggestion, which allowed us to provide additional information about our patient cohort. As showed in the table in the attached file (only for reviewer knowledge), the patients who do not respond seem to be younger and characterized by higher level of LDH, N/L ratio, and ECOG than the responders. These findings are in line with the results of the univariate analysis. Specifically, in the revised version of the manuscript, we have graphically depicted the level of normalized LDH according to response (DC, DP) and cohort (training, validation) (Figure 1, Panel b).  As reported in the manuscript, the predictive capability of normalized LDH to discriminate responders from no responders increases when it is included in a multivariate model together with age and ECOG (difference: -0.0585; p-value: 0.0298; 95%CI: -0.111; -0.0057).

Abbreviations such as DC, PD, ECOG, etc should be spelled at their first use. Especially, what is ECOG and why is it an important parameter? Authors' method has three parameters and one of them is ECOG however no where the authors have defined the meaning and significance of ECOG.:

-       We have reviewed and corrected all abbreviations, spelling them at the first use.

-       The definition of ECOG was already present in the first version of the manuscript (Result section line 130): “The Performance Status was defined according to the Eastern Cooperative Oncology Group (ECOG) criteria [ref. 30]. The ECOG score variable was dichotomized into two groups: 0 and ≥1.”

ROC has not been defined/spelled:

We have defined the meaning of “ROC” in the text (Results section).  “The predictive capability of the multivariate model was evaluated by generating a receive operating characteristic curve (ROC) and using as pivotal statistic the area under the ROC curve (AUC).”

In the study population were the values used for LDH and ECOG were recorded before the patients were given ICIs or after or during the treatment. And who would that affect the conclusion made in the manuscript.

All the variables analyzed are considered before starting ICIs, according to the purpose of the study, which is identifying useful parameters to select patients before starting therapy.

Reviewer 3 Report

The paper by Cona et al provides a preliminary assessment of the correlation of LDH, age, and ECOG score with response to immune-checkpoint inhibitors in a retrospective study of 271 clinical samples.  The paper is of interest as a preparatory study, but it would be beneficial to the reader if the paper was more organised.  Notably the authors should address the following:

Define all acronyms at first usage – the paper is littered with acronyms that are not defined or only defined after first usage.  Secondly, avoid unrecognised abbreviations – the authors use pts for patients – an abbreviation more recognised for points rather than patients, and furthermore they use this in some places but not others, for example once but not in both instances in line 144.  The authors also use disease control (DC) and for progression of disease (PD) – would it not be more logical to have disease progression (DP)?   

Although fitting with the journal format that the methods section is written last, a reader has to search the methods to have any idea about the cohort used, the authors provide no introduction to that at all in the results.  At least explain to the reader what you are doing and why – the reader is referred to Figure 1 without any further details, and not least, there is no Figure legend present to make interpretation and understanding clearer.  Provide Figure Legends, and order the manuscript appropriately.  For example, Table 1 provides the details of the clinical analysis and definitions of many of the acronyms already used in line 55, but this text should follow an introduction of the clinical samples.

Figure 2 – either alter to use darker colours or change the format of the lines themselves as it is very difficult to distinguish the line colours.

Discussion lines 95-97 discusses the need for predictive biomarkers, but here or elsewhere there is no discussion of the pharmacogenomics of the patients.  Could any specific genomic profiling provide an indicator of responsiveness to treatment, has this been assessed?

The Discussion section does not contain any details that relate to the possible mechanism associated with increased serum LDH activity.  Why would serum activity change? LDH protein release is often used as a marker of loss of cell membrane integrity and cell death, but herein the authors focus on LDH activity.  The authors mention the Warburg effect but this is within tumorigenic cells, not LDH floating around in the serum.   It would be useful to provide some mechanistic insight as to why this would be useful.

Lastly, in terms of limitations, the study numbers are small, and as mentioned above there is no mechanistic or genomic basis linked to responders.

Author Response

Define all acronyms at first usage – the paper is littered with acronyms that are not defined or only defined after first usage.  Secondly, avoid unrecognised abbreviations – the authors use pts for patients – an abbreviation more recognised for points rather than patients, and furthermore they use this in some places but not others, for example once but not in both instances in line 144.  The authors also use disease control (DC) and for progression of disease (PD) – would it not be more logical to have disease progression (DP)? 

According to your suggestion, we have reviewed and corrected all abbreviations, spelling them at the first use.   

Although fitting with the journal format that the methods section is written last, a reader has to search the methods to have any idea about the cohort used, the authors provide no introduction to that at all in the results. 

We have introduced the result section with more details about population.

At least explain to the reader what you are doing and why – the reader is referred to Figure 1 without any further details, and not least, there is no Figure legend present to make interpretation and understanding clearer.  Provide Figure Legends, and order the manuscript appropriately.  For example, Table 1 provides the details of the clinical analysis and definitions of many of the acronyms already used in line 55, but this text should follow an introduction of the clinical samples. Figure 2 – either alter to use darker colours or change the format of the lines themselves as it is very difficult to distinguish the line colours.

We have implemented figures legend and improved the graphics of figure as requested.

Discussion lines 95-97 discusses the need for predictive biomarkers, but here or elsewhere there is no discussion of the pharmacogenomics of the patients.  Could any specific genomic profiling provide an indicator of responsiveness to treatment, has this been assessed?

This is a very important point, but unfortunately we have no data about genomic profile of the patients. In the ongoing prospective study, we have planned to perform genomic profile of the patients to integrate and to validate our method.

The Discussion section does not contain any details that relate to the possible mechanism associated with increased serum LDH activity.  Why would serum activity change? LDH protein release is often used as a marker of loss of cell membrane integrity and cell death, but herein the authors focus on LDH activity.  The authors mention the Warburg effect but this is within tumorigenic cells, not LDH floating around in the serum.   It would be useful to provide some mechanistic insight as to why this would be useful.

Serum LDH levels derive from the lysis of cancer cells and the consequent release of the LDH enzyme into the blood. However, these levels also depend on LDH concentration inside tumor cells, which, in turn, reflects the glycolytic activity of the tumor. Indeed, LDH catalyzes the last step of aerobic glycolysis, i.e. the conversion of pyruvate to lactate. Therefore, serum LDH levels derived from dead cells are a function of glycolytic activity of neoplastic cells and of tumor burden. However, this does not explain why serum LDH should be associated with clinical benefit from ICIs. Of note, recent preclinical studies have demonstrated that inhibiting glycolysis in preclinical tumor models is associated with reduced tumor infiltration by immunosuppressor myeloid cells, with enhanced infiltration by cytotoxic lymphocytes and with improved anticancer activity of immune checkpoint inhibitors. Therefore, tumor glycolysis, may hamper antitumor immunity, thus contributing to make tumors resistant to immunotherapy. The association between high tumor glycolytic activity and resistance to ICIs could be explained by a competition model, according to which cancer cells and lymphocytes compete for the same molecules of glucose in TME. Once glucose is depleted by highly glycolytic, LDH overexpressing cancer cells, lymphocytes are unable to proliferate and undergo activation; in these metabolic conditions, the anticancer activity of ICIs, which requires lymphocyte activation, could be severely impaired.

An alternative explanation to the link between tumor glycolytic activity and response to immunotherapy comes from a recently published study (Li Wei et al. Cell Metabolism 2018; 28, 87-103), where inhibition of glycolysis in cancer cells was associated with reduced secretion of G-CSF and GM-CSF by cancer cells, with a consequent reduction of intratumor MDSCs, which are known to inhibit the activity of cytotoxic lymphocytes. Therefore, high serum LDH, which reflect tumor glycolytic activity, may also reflect a more immunosuppressive, MDSC-enriched tumor microenvironment.

Lastly, in terms of limitations, the study numbers are small, and as mentioned above there is no mechanistic or genomic basis linked to responders.

This is a proof of concept study whose interesting results need a larger series of patients with omogeneous diseases to be confirmed. In the ongoing prospective study, we planned to also evaluate the patient’s genomic characteristcs.

Reviewer 4 Report

The Manuscript entitled “Combination of Baseline LDH, Performance Status and Age as Integrated Algorithm to Identify Solid Tumor Patients with Higher Probability of Response to Anti PD1 and PDL1 Monoclonal Antibodies” submitted by Cona et. al., contains interesting results on the predictive biomarkers to immune-checkpoint inhibitors (ICIs). A major goal of personalized medicine in oncology is the identification of drugs with predictable efficacy based on a specific trait of the patients. In this way, the results claimed in this manuscript are worth publishing in the journal “Cancers”

Author Response

No particular issue to be addressed

Round 2

Reviewer 2 Report

Authors have made significant improvements in the revised version of the manuscript. The manuscript could be considered for publication.